# Unlocking the Complexity of Neuromuscular Diseases: Insights from Human Pluripotent Stem Cell-Derived Neuromuscular Junctions

**DOI:** 10.3390/ijms242015291

**Published:** 2023-10-18

**Authors:** Morgan Gazzola, Cécile Martinat

**Affiliations:** INSERM U861, Institute for Stem Cell Therapy and Exploration of Monogenic Diseases, 91100 Corbeil-Essonnes, France; cmartinat@istem.fr

**Keywords:** pluripotent stem cells, human neuromuscular junction, in vitro models, microfluidics and organoids

## Abstract

Over the past 20 years, the use of pluripotent stem cells to mimic the complexities of the human neuromuscular junction has received much attention. Deciphering the key mechanisms underlying the establishment and maturation of this complex synapse has been driven by the dual goals of addressing developmental questions and gaining insight into neuromuscular disorders. This review aims to summarise the evolution and sophistication of in vitro neuromuscular junction models developed from the first differentiation of human embryonic stem cells into motor neurons to recent neuromuscular organoids. We also discuss the potential offered by these models to decipher different neuromuscular diseases characterised by defects in the presynaptic compartment, the neuromuscular junction, and the postsynaptic compartment. Finally, we discuss the emerging field that considers the use of these techniques in drug screening assay and the challenges they will face in the future.

## 1. Introduction

Neuromuscular disorders (NMDs) are a large group of genetic diseases that affect or progressively block the control and strength of voluntary muscle movement. It is estimated that these diseases affect about 1 in 1000 individuals in Europe [1]. The signature symptoms associated with NMDs are progressive muscle atrophy and weakness, resulting in mild to severe symptoms with elevated lethality. The onset of these pathologies is variable, ranging from birth to adulthood. More than 1000 NMDs have been identified so far and can be broadly divided into different groups: motor neuron (MN) disorders, such as spinal muscular atrophy; neuromuscular junction (NMJ) disorders, such as congenital myasthenic syndromes; and skeletal muscle disorders, such as muscular dystrophies and other myopathies. The identification of over 600 causative genes highlights the genetic variability of NMD [2]. However, many genes remain to be discovered, and several thousands of patients with NMDs are still currently without a definite diagnosis in Europe. In addition, some NMDs such as myasthenia gravis are not due to genetic variations but rather to autoimmune disorders. The heterogeneity of these diseases, combined with the low prevalence of most of the diseases, has made it challenging to determine the underlying causes and to develop targeted treatments. In addition, treatment options for most of these diseases are extremely poor, often limited to supportive care that does not prevent disease progression.

Several challenges may explain the lack of progress in identifying treatments for NMDs. First, the rare biopsies available for these diseases are almost exclusively muscular and therefore do not show the complexity of the neuromuscular system, in particular the NMJ, which plays a crucial role in transmitting signals from the nervous system to muscles. In addition, it is extremely difficult to assess disease progression. Sampling at presymptomatic stages is extremely rare, and multiple sampling at different time points after diagnosis is difficult, if not impossible. Other major limitations include the long delays in diagnosing patients, as well as the lack of reliable biomarkers to detect disease onset.

To overcome these difficulties, many animal models have been developed in different species, including invertebrate models such as drosophila or *C. elegans*, and small vertebrate models such as rodents or zebrafish. In most cases, these animal models are poor phenocopies of human disease due to genetic and physiological differences in the neuromuscular system between humans and other species [3,4,5,6]. For example, human NMJs are significantly more fragmented and have the smallest nerve terminal surface area currently known in vertebrates. As a result, a smaller amount of the neurotransmitter acetylcholine (ACh) is released per action potential. However, human NMJs have deeper postsynaptic folds than mouse NMJs, which contribute to the amplification of ACh signalling [3,6,7,8,9]. Several differences in the molecular pathways have also been identified [4]. To illustrate the differences between human and animal models, spinal muscular atrophy, one of the most common genetic disorders affecting young children, is associated with mutations in the *SMN1* gene. The human genome is the only one that contains two *SMN* genes (*SMN1* and *SMN2*), whereas all species used to model spinal muscular atrophy (SMA) have only one *SMN* gene equivalent to *SMN1*. In all organisms except humans, the loss of the *SMN* gene results in embryonic lethality, leading to the development of complex genetic models to overcome this limitation. In addition, animal models are often engineered to delete the causative gene, if known, rather than reproduce the human mutations, which can confound pathology studies. Finally, the use of animal models is poorly compatible with large drug discovery campaigns and does not comply with the current restrictions on compound testing in animal models.

Recent advances in human pluripotent stem cells have revolutionised the neuromuscular field. Even complex units such as the NMJ can now be produced in vitro so that each component of the motor unit can be studied in isolation or in interaction with others. This domain has been extensively reviewed in several publications [10,11]. In parallel with technological breakthroughs, these developments have also started to transform the field of NMD modelling. Here, we review the various human pluripotent stem cell-based models that have been investigated to improve our understanding of normal and pathological NMJ development. We also review the current treatments already approved by the Food and Drug Administration (FDA) for five paradigmatic NMDs, and we highlight the three molecules discovered in hIPS-derived motor neurons or muscles that are currently in clinical trials. Finally, we conclude with the challenges and strategies to overcome the limitations of these models that would allow for predictive, patient-specific screening studies and future potential NMD treatments.

## 2. In Vitro NMJ Models: From Two-Dimensional Co-Culture to Three-Dimensional Neuromuscular Organoids

### 2.1. Two-Dimensional Co-Culture

The pioneering integration of a presynaptic compartment derived from human embryonic stem (hES) cells was accomplished by Li and collaborators in 2005 [12]. These protocols were derived from previously established protocols for deriving mouse embryonic stem (mES) cells into MNs [13,14]. Although the derivation of hES cells did not result in pure MN populations, this proof of concept marked a remarkable advance in in vitro NMJ modelling. The culture of hES-derived MNs over a monolayer of mice C2C12 myoblasts allowed for the observation of ACh receptors (AChR) clusters juxtaposed to neurites after three weeks of culture. Subsequently, numerous hybrid co-culture systems were established by pairing hES-derived MNs with rodent-derived myotubes [14,15,16,17]. The first in vitro model of a fully human NMJ, with both the human pre- and postsynaptic compartments, was achieved by Marteyn and colleagues in 2011 [18]. MNs were derived from hES cells by adapting established protocols [12] and co-cultured with primary human skeletal muscle cells. Such co-cultures have also been performed with MNs derived from spinal cord stem cells [19,20]. Within these fully human co-culture systems, the assessment of endplate potentials and contractions provided compelling evidence for robust NMJ functionality [20].

The discovery of human induced pluripotent stem (hiPS) cells by Takahashi and Yamanaka in 2006 [21] revolutionised the field of pluripotent stem cells. Obtaining these cells through the dedifferentiation of human cells has overcome the ethical problems associated with the use of ES cells. In addition, the discovery of hiPS cells has made it easier for different laboratories to obtain pluripotent stem cells, thus democratising their use. Five years after their discovery, the first co-cultures of primary rodent skeletal muscle with hiPS-derived MNs were developed by Stockmann and colleagues in 2011 [22] by adapting the protocols described in 2005 [12]. By employing hiPS cells derived from keratinocyte dedifferentiation, the authors successfully re-differentiated these cells into MNs over a 28-day period [22]. Using electron microscopy, the authors discerned organised synaptic vesicles within the axonal terminal region, closely associated with myotubes. However, they were unable to observe the three-dimensional structure of mature NMJs, such as the postsynaptic folds.

From a functional point of view, a co-culture study was carried out between hiPS-derived MNs and primary chick muscle tissue [23]. The co-cultures resulted in the establishment of anatomically mature and functional NMJs, as evidenced by the restoration of the contractile function of chick myofibres. Emblematic pretzel-shaped endplates emerged after five weeks of co-culture, and endplate potentials sensitive to tubocurarine indicated functional cholinergic communication between the two synaptic compartments. However, even if functional, no ultrastructural studies have been performed to assess the presence of postsynaptic folds [23]. It is important to note that the need for the presence of folds in the NMJ has yet to be investigated [24].

An important step has been taken with the initiation of studies using identical donor cell lines to generate both MNs and skeletal muscle, paving the way for the comprehensive modelling of NMJs from a single individual [25,26,27]. Functional NMJs were also successfully established in these studies. Unfortunately, the visualisation of three-dimensional morphological structures via ultrastructural analysis remained elusive, even after more than 100 days in co-culture [27]. Remarkably, in 2020, Mazaleyrat and colleagues were able to co-differentiate both MNs and myotubes from the same hiPS cell culture within 19 days. This was achieved through the targeted use of bone morphogenic proteins (BMP) inhibitors, CHIR, insulin, and hepatocyte growth factors [28]. Using ultrastructural characterisation, they revealed a distinct skeletal Z-disc intercalation accompanied by the presence of juxtaposed synaptic buttons over muscle fibres. By assessing muscle contractions and calcium flux within the postsynaptic compartment, they also confirmed the functionality of their neuromuscular system [28].

Subsequently, the optimal microenvironment for the development of neuromuscular synapses has been the subject of extensive research [29]. Using polyacrylamide hydrogels, Takahashi and colleagues designed micropatterns of ridges and grooves to facilitate the development of pseudotissues that resemble human muscle fibre sheets [30]. Using a fibrin gel on their culture, the authors were able to generate precisely aligned human myotube fibres differentiated from primary myoblasts. By incorporating hiPS-derived MNs into the hydrogels, they were able to quantify tissue contraction, which was abolished by the addition of curare.

A major limitation of two-dimensional models is their ability to recapitulate the entire NMJ niche, including terminal Schwann cells (tSCs) and kranocytes [31,32]. The presence of tSCs is, therefore, necessary for synapse development, maintenance, and plasticity. In 2021, the first establishment of an in vitro tri-culture model comprising muscle, MNs, and Schwann cells was realised [33]. The MNs and Schwann cells were both generated through the differentiation of hiPS cells and sequentially introduced on a layer of murine myoblasts. Notably, the authors observed an increase in AChR cluster size in myotubes following the sole introduction of Schwann cells. No discernible changes in the number or size of AChR were observed between the simple MNs and muscle co-culture and the tri-culture system [33]. The development of such a functional framework provides an insightful approach to assessing the impact of Schwann cell addition in co-culture models. The incorporation of other components of the neuromuscular niche, such as glial or Pax7+ cells, could also be proposed in future investigations.

In all of these culture systems, even when direct interfaces were established between MNs and muscle cells, a significant hurdle remained in assessing the NMJ activity. Using voltage-clamp recordings, researchers were limited to monitoring either the pre- or the postsynaptic compartment. Over the years, new approaches have overcome this limitation by combining novel methods. Fluorescence microscopy and dual-patch clamp allowed for the simultaneous measurement of NMJ activity in both the pre- and postsynaptic compartments [34]. By integrating optogenetic tools into their systems, researchers succeeded in taking control of the NMJ by controlling MN firing [35].

### 2.2. Microfluidics

Over the past decade, the development of microfluidic systems has aimed to improve compartmentalisation between pre- and postsynaptic compartments, providing precise spatial and temporal control of cellular microenvironments. The first microfluidic chips in the neuromuscular field were primarily designed to facilitate axonal growth within microtunnels, limiting the number of neurites per channel and restricting the location of axonal bodies [36]. The pioneering establishment of neuromuscular interactions in a microfluidic device using mouse embryonic stem cells was achieved by Park and colleagues in 2013 [37]. Mouse ES-derived MNs extended through microfluidic tunnels, establishing direct contact with C2C12 myotubes, as evidenced by α-bungarotoxin (α-BTX) labelling. Subsequently, the formation of functional NMJs within a microfluidic chamber was realised three years later in 2016 [38]. Using mES-derived MNs and embedding mice C2C12 myoblasts within a 3D hydrogel, the authors simulated the physical separation between the two synaptic compartments. The introduction of compliant columns directly integrated into the muscle compartments has provided a versatile in vitro system for dynamic and quantitative force assessment [38]. The addition of electrodes to the muscle compartment of microfluidic chips has further enhanced the ability to interrogate the NMJ using either drug treatments or electrical stimulation [39]. Moreover, the ability of these systems to generate mid-throughput NMJ platforms was investigated by using both MNs and myotubes differentiated from the same hIPS cell lines [40]. Interestingly, by combining microfluidic chambers and optogenetics, Machado and colleagues demonstrated the positive effect of entrainment on the NMJ formation [41]. Mouse ES-derived MNs expressing channelrhodopsin 2 (ChR2) were trained with 20 ms of blue light exposure at 5 Hz, 1 h per day for 5 days. After 5 days of training, the authors observed improved myofibre innervation and an increased number of neuromuscular synapses formed [41]. In an interesting study, by integrating skeletal muscle tissue and hES-derived MNs expressing ChR2 into a soft scaffold, a biohybrid machine was established, named “flagellar swimmers” [42]. By generating cyclic light exposure, the authors were able to activate the muscle contraction that triggers the swimmer’s locomotion. Finally, a reproducible method to generate NMJs using human-derived iPS in commercial microfluidic chips has recently been described [43,44]. The same system has also been described to establish a tri-culture model between hiPS-derived MNs, hiPS-derived astrocytes, and primary skeletal muscle [45].

Having compartmentalised different cell types, the challenge remained to recapitulate cell interactions in a more relevant microenvironment reflecting the three-dimensionality of in vivo tissues. Thus, the generalisation of hydrogels allowed for the development of such models. Following the pioneering work combining three-dimensional hydrogels in microfluidic devices [38], 3D NMJ models were performed by seeding human primary myoblasts in hydrogels composed of fibrin and Geltrex [46]. When hES-derived MN spheres were added to these three-dimensional structures and stimulated with L-glutamate, MN exocytosis, calcium transients, and synchronous tissue contractions were observed [46,47]. Combining microfabrication, engineering, optogenetics, and image processing, the first semi-automated human NMJs were developed, paving the way for using 3D NMJs in high-throughput screening [48,49]. Finally, a well-described protocol for generating a human physiological NMJ platform in a three-chamber microfluidic device has recently been published [50].

### 2.3. Bioprinting

Recently, the emerging field of bioprinting has provided new opportunities for precise myofibre patterning. The first robust methods for encapsulating and aligning myoblasts in a three-dimensional system were performed using bioprinted silk–fibrin cantilevers [51]. The authors used their system as a culture support to add MNs derived from human induced neuronal cells. Following the addition of L-glutamic acid, the authors were able to measure calcium flux through artificially aligned myoblasts. Bioprinting could therefore be a good way to improve the orientation of muscle fibres in culture. Kim and colleagues highlighted the feasibility of using bioprinted 3D hydrogel scaffolds to induce muscle fibre alignment in in vitro artificial muscles. Such scaffolds support the culture, differentiation, and alignment of primary human skeletal muscle cells. Interestingly, this technique appears to be compatible with the functional integration of immortalised human neural progenitor cells [52]. The bioprinted constructs with neural cells facilitate rapid innervation and maturation into organised muscle tissue, restoring normal muscle function in a rodent model of muscle defect injury [52]. Another interesting application of bioprinting is the reproduction of the cellular microenvironment with extracellular matrix-mimicking bioinks. These inks will provide microenvironmental molecules and rheology to hiPS-derived MNs [53]. By removing the cells from porcine spinal cords, and solubilising the remaining tissues, Kong and colleagues provided a bioink for the long-term maintenance of their culture. Such a bioink could also be used for in vivo applications such as cell transplantation or spinal cord injury [53]. Similarly, by generating an optimal hydrogel containing methacrylated gelatine, collagen, and decellularised extracellular matrix, the bioprinted methacrylate structure showed interesting in vivo regenerative potential [54]. The authors observed a significant increase in the regeneration of muscle transplanted with scaffolds compared with controls, with no topographical alterations. These developments should be pursued in the future to improve the maturation of NMJ systems in vitro and highlight an area of considerable interest in the field of tissue engineering.

### 2.4. Organoids

A major breakthrough in stem cell research is the emergence of new three-dimensional models called organoids. These complex structures derived from the differentiation of pluripotent stem cells, demonstrate the ability to recapitulate native and functional tissues in vitro [55]. A key advantage of these models is their ability to generate multiple cell types in self-assembling three-dimensional arrangement [55]. In addition, these small organs are also more physiologically relevant than two- or three-dimensional co-cultures. However, although the scalability of organoids appears robust enough for drug screening, their reproducibility through multiple differentiation remains an open question.

Recent major discoveries in the neuromuscular field have paved the way for future investigations. In 2020, Martins and colleagues generated the first self-organised neuromuscular organoids from human pluripotent stem cells [56]. By differentiating human ES cells into neuromesodermal progenitors (NMPs) and aggregating them into three-dimensional structures, the NMPs self-organised into well-defined neuronal and mesodermal regions. After 150 days, the authors described the presence of mature NMJs with the presence of terminal Schwann cells. More interestingly, the ultrastructural analysis revealed a three-dimensional NMJ organisation that includes immature synaptic folds in the postsynaptic region [56,57]. This organisation has never been described so far in co-culture systems. Another way to connect MNs to muscles using organoids is to connect them directly using the extracellular matrix as pseudohydrogels. This allowed Andersen and colleagues to assemble three different organoids to recapitulate the connection between cortical neurons, MNs, and skeletal muscle in a complex three-dimensional model [58]. They separately differentiated spinal and cortical organoids from hiPS cells and generated a muscle spheroid from primary human skeletal muscle cells. The three organoids were then incubated closely together in transwells for a few days. A primary connection between the cortical and spinal organoids was observed, as evidenced by the presence of neuronal projections that form functional synapses. In addition, the authors observed neuronal projections from spinal organoids to skeletal spheroids, leading to more synchronous contraction from the muscle region. Finally, by expressing the light-sensitive channel ChR2 in cortical organoids, the authors were able to measure a contractile response from skeletal muscles following light stimulation, demonstrating the connection between the three organoids [58]. By patterning NMP spheres, Pereira and colleagues differentiated sensorimotor organoids from hiPS cells in 2021 [59]. The authors generated neuromuscular organoids containing neuronal, glial, endothelial, and muscular regions using a relatively similar protocol to that described by Martin and colleagues [56,59]. Using optogenetics, the authors were also able to observe functional NMJs in their tissue. Finally, recent work on neuromuscular organoids has focused on the muscle region of such organoids and their intrinsic ability to regenerate after injury [60]. By differentiating hESCs into 3D human skeletal muscle organoids (hSKMOs), the authors recapitulated the myogenesis process. Interestingly, these organoids demonstrate the regenerative capacity of muscle in vivo due to the presence of PAX7+ cells (satellite cells). After muscle injury with cardiotoxin, hSKMOs showed myofibre degeneration followed by a regeneration process [60]. Such organoids represent interesting models to study the consequences of tissue inflammation on muscle regeneration.

In summary, the generation of human NMJs in vitro by differentiating one or more components from pluripotent stem cells is a fascinating field that has evolved drastically over the last 20 years. From two-dimensional cell co-cultures to three-dimensional complex assembloids (Figure 1), a wide range of models has been developed, each with its strengths and weaknesses. Interestingly, the protocols developed allow for the detection of different levels of NMJ maturity, apparently only detectable through ultrastructural studies. Another interesting point will be to evaluate the need to vascularise such systems. Is the presence of endothelial cells necessary for the enhancement of NMJ maturation and organisation? In addition, some studies have demonstrated the presence of sympathetic neurons in close proximity to the NMJ [61]. The development of such models will bring us even closer to the NMJ that can be observed in vivo. Despite the exciting development of such models, their use in disease modelling requires further investigation to enable the development of screening assays.

## 3. Co-Culture and Three-Dimensional Models for Studying Neuromuscular Disorders

Over the years, scientists have screened several molecules that have actually been approved by the FDA for the treatment of NMDs (Table 1). The establishment of artificial NMJs in vitro further enhances the potential to investigate the intricate mechanisms underlying the progression of these diseases. These approaches provide dynamic platforms to dissect the multifaceted interactions between neural and muscular components, ultimately advancing our understanding of these disorders at the molecular, cellular, and functional levels. In this section, we will see how the systems described above have been used to understand paradigmatic diseases affecting the presynaptic compartment, the neuromuscular junction, or the postsynaptic compartment.

### 3.1. Amyotrophic Lateral Sclerosis

Amyotrophic lateral sclerosis (ALS) is a devastating neurodegenerative disease characterised by progressive degeneration of the upper and lower MNs, leading to muscle weakness, paralysis, and death [62]. This pathology is directly linked to genetic mutations in the *c9orf72*, *tar DNA binding protein-43* (*TDP-43*), *fused in sarcoma* (*FUS*) and *superoxide dismutase 1* (*SOD1*) genes [62]. The consequences of these mutations on the NMJ have been investigated in different animal models (for a recent review, see [63]). In a mouse model of *SOD1* mutations, axonal distal regions and NMJ integrity were shown to be affected before the onset of clinical symptoms [64]. In addition, in ultrastructural studies, d mitochondrial disorganisation in the nerve terminal and a reduced number of synaptic vesicles were observed [65,66]. In zebrafish, the expression of the mutant *SOD1* gene resulted in abnormal motor neuron branching and the defective formation of presynaptic nerve endings [67]. In a series of comprehensive investigations, researchers have used co-culture experiments to shed light on the complex mechanisms underlying ALS pathology [41,43,45,49,68,69,70,71,72,73]. The use of co-culture models makes it possible to study the effects of a diseased presynaptic compartment on healthy muscles, or vice versa.

To understand the consequences of mutant forms of *TDP-43* and *FUS* in ALS, Wächter and colleagues set up an experiment in which they overexpressed both wild-type and mutant human isoforms in a murine ES cell line [68]. To investigate the non-cell-autonomous effects of mutant protein overexpression, they differentiated ES cells into skeletal muscle and cultured them with healthy MNs. While the number of MNs remained unchanged after 24 h of culture, a significant reduction in MN neurite lengths was observed for MNs cultured with muscle cells expressing the mutant proteins *FUS* and *TDP-43* [68]. Through this conceptual study, the authors highlighted the detrimental effects of mutant *TDP-43* and *FUS* on the surrounding environment. The consequences of *TDP-43* mutations were also investigated using a three-dimensional co-culture system with ALS patient-derived MNs in a microfluidic device [49]. The authors revealed a cascade of consequences, including a reduction in Islet1-positive cells during MN differentiation, a reduced neurite outgrowth rate, and a decrease in muscle contraction frequency. Strikingly, over a 14-day co-culture period, muscles interacting with ALS MNs showed increased caspase 3/7 activity. Through their deleterious effects on both MNs and muscles, these two studies demonstrated that *TDP-43* and *FUS* mutations have anterograde and retrograde effects on MNs.

Shifting the focus toward the NMJ, microfluidic devices probed the influence of the *FUS* mutation on the NMJ formation [43,69]. Reduced NMJ numbers and altered NMJ morphology were observed in co-cultures with hiPS-derived MNs from ALS patients. Using a targeted approach, treatment with the histone deacetylase 6 (HDAC6) inhibitor tubastatin A successfully restored impaired neurite outgrowth and NMJ numbers [43]. Semi-automated mid-throughput quantitative analysis in co-cultures with hiPS-derived compartments from juvenile ALS patients or controls revealed impaired endplate maturation in *FUS*-ALS myotubes compared with controls. Additional perturbations were observed in the expression level of the alpha 1 subunit of the AChR in affected myotubes [69]. A novel perspective emerged as *FUS*-ALS hiPSC-derived astrocytes were found to exert a dual impact on human motor units, involving gain-of-toxicity and loss-of-support mechanisms [45]. The researchers used iPS cells from *FUS*-ALS patients to derive astrocytes and MNs and cultured them in a microfluidic environment with healthy myotubes from human biopsies. Interestingly, ALS-derived astrocytes exhibited a cytotoxic effect, disrupting MN neurite outgrowth, NMJ formation, and overall functionality [45]. It is interesting to note that these results are consistent with recent studies demonstrating the pro-inflammatory reactive state of astrocytes in ALS [74,75]. In 2020, Guo and colleagues established clinically relevant parameters for the NMJ state in a microfluidic system using ALS patient-derived MNs [70]. MNs from ALS patients, including those with *SOD1* and *FUS* mutations, showed deterioration in parameters such as NMJ number, fidelity, and fatigue index. In their study, the authors evaluated the holistic Deanna Protocol (DP), a protocol that includes supplements that target a spectrum of cellular mechanisms, such as facilitating cellular metabolism and reducing oxidative stress. This protocol demonstrated positive effects by rescuing each functional mutant phenotype.

Common defects associated with ALS also include the mutations in the *SOD1* gene, where MN decay appears to be linked to the interactions with astrocytes expressing a mutant form of the *SOD1* gene [76]. Compartmentalised co-culture in microfluidic devices showed that interactions between mES-derived MNs and astrocytes expressing the mutant form of *SOD1* (*SOD1G93A*) reduced MN survival, significantly impaired MN–muscle connectivity, and led to myofibre displacement [41]. Notably, these phenotypes appear to be reversed by treatment with the RIPK1 inhibitor necrostatin [41]. Badu-Mensah and colleagues used a microfluidic chip to study the effects of two different *SOD1* mutations, *L144P* and *E100G*, on the NMJ [72]. They differentiated both WT and *SOD1*-mutant hiPS cells in skeletal muscles and MNs and generated four co-cultures consisting of WT skeletal muscles (WT MNs), *SOD1* skeletal muscles (WT MNs), WT skeletal muscles (*SOD1* MNs), and *SOD1* skeletal muscles (*SOD1* MNs). By crossing such experimental conditions, they found the greatest decrease in NMJ number and stability induced by *SOD1* mutations affecting muscles, highlighting muscle deficiency as the main factor [72]. In addition, a 3D cell culture system for bioengineering human NMJs to model ALS showed a slight reduction in muscle displacement when neuromuscular junctions were established with MNs derived from ALS hiPSCs [73].

Finally, using neuromuscular organoids, Pereira and colleagues embarked on a comprehensive investigation involving both hiPS-derived ALS patient-derived cells and isogenic cell lines generated for *TDP43*, *SOD1*, and *PFN1* mutations [59]. Over a period of 7 weeks, the authors observed a significant reduction in large contractions in organoids derived from ALS patient hiPS cells compared with controls. Using an isogenic approach, the researchers carefully analysed the effects of each mutation on muscle innervation in their model. Interestingly, they observed a decrease in the percentage of muscle innervation in cell lines with *SOD1* and *PFN1* mutations, whereas no discernible difference was observed in *TDP43*-mutant hiPS cells [59]. Remarkably, the area of innervated neuromuscular junctions (NMJs) showed a decrease in *TDP43* mutant cells but not in cells with *SOD1* and *PFN1* mutations.

Collectively, by mimicking the NMJ affected by ALS pathology separately in the pre- and postsynaptic compartments, as well as in the whole NMJ, these studies have highlighted novel cellular and molecular mechanisms involved in this disease. Therefore, the use of co-culture and three-dimensional organoids represent interesting models to unravel the pathological mechanisms involved in NMDs.

### 3.2. Spinal Muscular Atrophy

Spinal muscular atrophy (SMA) type 1 is characterised by the severe degeneration of the spinal cord α-MNs, leading to muscle weakness, muscle atrophy, and patient death [77]. This disease caused by mutations in the survival motor neuron 1 (*SMN1*) gene, encompasses a range of severity and affects both children and adults. SMA has gained considerable attention due to its impact on motor function and quality of life and has inspired rigorous research efforts to elucidate its underlying mechanisms and develop innovative therapeutic options [77]. To elucidate the pathogenesis of SMA, multiple investigations have established co-culture models utilising hiPS cell lines derived from SMA patients to generate MNs. In these systems, despite extended culture durations, the loss of MNs was not obvious, and a comparable number of HB9+ cells were present in cultures containing either wild-type or SMA patient-derived MNs [15]. However, a notable reduction in the capacity of SMA-derived MNs to generate expansive AChR cluster areas on co-cultured myofibres was observed after 40, 50, or 60 days of culture. Remarkably, treatment with valproic acid or a specific oligonucleotide targeting the *SMN2* exon 7 inclusion restored the ability of SMA-derived MNs to form normal levels of AChR clusters [15]. Using four SMA patient-specific hIPS cell lines, Boza-Moran and co-workers did not find reduced AChR cluster sizes after three days of co-culture with C2C12 myoblasts. Nevertheless, the lack of phenotype in their study may be due to the very short culture duration [78].

Surprisingly, these observations are different from studies generated with hiPS-derived MNs from SMA patients [79,80,81]. In their study, the authors observed a 50% loss of survival of MNs after 8 weeks in culture. However, these studies were carried out in simple cell cultures, without the presence of muscle cells.

An alternative approach to assessing the impact of SMA on NMJ formation is to artificially knock down the *SMN1* gene. Lin and colleagues used MNs and myotubes, both differentiated from hiPSCs knocked down for the *SMN1* gene. The authors observed reduced maximal contraction velocity and contractile synchrony compared with controls [27]. Additionally, unexpected fragility of the myotubes was observed, revealing the cellular structural vulnerability caused by the *SMN1* mutation. Ultrastructural analysis revealed reduced mitochondrial biogenesis with disruptions in the muscle sarcomeric organisation [27].

Given the inaccessibility of MN biopsies from healthy humans and SMA patients, the differentiation of hiPS cells into MNs to model the NMJ will allow the study of cellular and molecular mechanisms associated with this pathology. Furthermore, the discovery of a protocol for the differentiation of spinal and cranial MNs in 2015 will allow for the study of selective MN vulnerability in SMA [82].

### 3.3. Myasthenia Gravis

Myasthenia gravis (MG) is a chronic autoimmune neuromuscular disease that affects nerve–muscle communication characterised by muscle weakness and fatigue. MG occurs when the immune system mistakenly targets and attacks AChRs on the surface of muscle cells, hence affecting the good NMJ function [83]. A different form of MG is known as the congenital form of the disease [84]. In these cases, the pathology can be caused by 32 identified genes that affect the presynaptic, synaptic, and postsynaptic compartments [84]. Recently, patient-derived models have been generated using MNs derived from hiPSCs [85]. Interestingly, the authors observed an accumulation of agrin in the sarcoplasmic reticulum of patient-derived cells without altering their ability to trigger the formation of AChR clusters [85]. In both types of MG, signals from nerves to muscles are disrupted, leading to difficulties in controlling voluntary muscle movements. Several models have been used to assess the effects of MG syndromes on NMJs in vitro, from two-dimensional co-culture systems to, more recently, three-dimensional organoids.

To model congenital MG, the use of Waglerin-1 (WTX) peptide, a small peptide that selectively binds and blocks the α subunit of the muscle AChR, has been described [46]. By adding WTX to a three-dimensional co-culture model of the NMJ, the authors measured a decrease in calcium and contractile activity compared with untreated neuromuscular cultures [46]. Similarly, using anti-nAChR antibodies in a microfluidic device containing hiPS-differentiated MNs and primary myotubes, Smith and colleagues reduced the NMJ stability, mimicking the clinical phenotype observed in MG [86].

Another method for reproducing the phenotype of MG is to directly use antibodies isolated from MG patients. Steinbeck and colleagues incubated antibodies and complements from patients in a co-culture containing hES-derived MNs expressing ChR2 and primary human myoblasts [35]. The authors observed a large reduction in muscle twitch and calcium response to light stimulation when co-cultures were incubated with patient immunoglobulin G (IgG) for 3 days. The phenotype was reversed after 6 days following the washout of IgG. In a semi-automated recording system, Vila and colleagues also observed the complete abolition of muscle contraction following the incubation of their system with serum isolated from MG patients [48]. Finally, a recent study by Martins and colleagues with neuromuscular organoids demonstrated the effect of MG antibodies on NMJ activity [56]. The authors incubated 50-day-old neuromuscular organoids with antibodies from MG patients for 72 h. Following incubation, neuromuscular organoids showed a decrease in the number of NMJs, measured using α-BTX labelling, as well as a decrease in muscle contraction, attested as a decrease in spontaneous organoid displacement [56].

These conceptual studies open up new possibilities for MG patients. Currently, MG diagnosis is difficult and requires blood samples, nerve tests, scans, and sometimes edrophonium tests. By incubating freshly collected patient serum in two-dimensional co-culture models or three-dimensional organoids, we can easily imagine using these methods as potential diagnostic techniques. However, the standardisation of large-scale production will remain an issue to be addressed.

### 3.4. Muscular Dystrophies

Myotonic dystrophy type 1 (DM1) and Duchenne muscular dystrophy (DMD) are two paradigmatic muscular dystrophies that primarily affect the muscles, causing progressive muscle weakness and deterioration over time.

DM1 is an autosomal dominant disease caused by the expansion of CTG nucleotide repeats within the *DMPK* gene, resulting in a variety of clinical manifestations that primarily affect the muscles and nervous system [87]. This pathology is characterised by progressive muscle weakness; myotonia (delayed muscle relaxation); and a wide range of systemic symptoms, including cardiac abnormalities, cognitive impairment, and endocrine disturbances [87]. This disorder represents a fascinating and complex interplay of genetic, molecular, and physiological mechanisms that continues to be the subject of extensive research to improve diagnosis, understanding, and potential therapeutic strategies. Few studies have investigated the effects of DM1 on NMJ formation using co-culture systems. By establishing the first fully human in vitro neuromuscular system, Marteyn and colleagues also assessed the consequences of DM1 pathology in their system [18]. The authors differentiated MNs from hES cells of DM1 patients and observed an increase in neuritogenesis, synaptogenesis, and neurite outgrowth compared with controls [18]. Interestingly, gene expression analysis revealed a decrease in the expression of two genes from the *SLITRK* family in DM1 MNs.

One of the most studied aspects of DM1 pathology is the sequestration of muscle blind-like proteins (*MBNL*) in nuclear foci due to the retention of toxic *DMPK* mRNA [88,89]. Using co-culture over micropatterns to normalise their measurements, Tahraoui-Bories and colleagues evaluated the consequences of *MBNL* depletion on NMJ maintenance and formation [90]. Human iPS cells that were knocked down for MBNL proteins and DM1 hiPS cells were differentiated into MNs and co-cultured with primary human myoblasts. Interestingly, the authors observed an increase in neurite outgrowth and calcium oscillation frequency in both DM1 and *MBNL* KO hiPS-derived MNs. They also observed a decrease in the size of AChR clusters in the postsynaptic compartment [90]. Over the years, researchers have focused on the muscular aspect of DM1 disease. Recent studies of this pathology are opening up new perspectives on the neuronal state of the disease. Although the “molecular players” are still to be explored, the focus is shifting to a more neuromuscular aspect.

Duchenne muscular dystrophy (DMD) is a severe, progressive, and fatal pathology directly linked to the mutation of the dystrophin gene on the X chromosome [91]. With an incidence of 1 in over 5000 boys worldwide, this disorder is the most common muscular dystrophy. Through their function of attaching cytoskeleton elements to cellular membranes, dystrophin mutations are known to cause membrane instability, altering cellular components such as the sarcolemma by making them leaky and susceptible to injury [91]. The consequence of DMD in NMJ formation and function was recently evaluated in two distinct studies using previously described models. Through the co-differentiation of hiPS cells from DMD patients, Mazaleyrat and colleagues reproduced NMJs with a diseased microenvironment [28]. In their system, using electronic microscopy, the authors observed a decrease in myofibre sizes as well as a vacuole-like area between the fibres. Altogether, these results point to the degeneration of muscle fibres [28]. In 2021, Paredes Redondo and colleagues aimed to study DMD consequences on NMJs using microfluidic chips containing skeletal muscle cells, MNs, and astrocytes derived from hiPS cells obtained from DMD patients [92]. Upon differentiation, pluripotent stem cells derived from DMD patients showed a significant decrease in the number of cells positive for pax7+, myoD+, and MF-20. Importantly, there was also a significant change in their ability to fuse into myotubes. Once in co-culture, DMD cells showed a decrease in contractile capacity as well as a large decrease in the number and area of AChRs. The authors were able to restore muscle contraction using a selective inhibitor of the transforming growth factor β (TGF-β pathway, SB-431542 [92].

Collectively, these studies emphasise the rationale for using pluripotent stem cell-derived NMJs to study neuromuscular diseases. Selecting specific compartments, co-cultures, and microfluidic systems allows for studying the consequences of a specific NMJ deficiency. Conversely, organoids can mimic a completely affected NMJ in its microenvironment. However, their use in the evaluation of new pharmacological compounds will be a challenge to address in the coming years.

**Table 1 ijms-24-15291-t001:** Current FDA-approved treatments for the five paradigmatic NMDs covered in this review. The green line represents the only treatment that was tested in hiPS-derived MNs during its development. For myotonic dystrophy type 1, there are no FDA-approved therapeutics.

Amyotrophic Lateral Sclerosis		
Compound name	Mechanism of action	Reference
Riluzole	A modulator of the glutamatergic transmission	[93]
Edaravone	Free radical scavenger	[94]
Sodium phenylbutyrate	Mitigating endoplasmic reticulum stress and mitochondrial dysfunction	[95]
Spinal Muscular Atrophy		
Compound name	Mechanism of action	Reference
Zolgensma	AAV9 vectors carrying a codon-optimised *SMN1* sequence	[96]
Spinraza	Oligonucleotide modifying the splicing of *SMN2*	[97]
Risdiplam	Oligonucleotide modifying the splicing of *SMN2*	[98]
Myasthenia Gravis		
Compound name	Mechanism of action	Reference
Tacrolimus	Macrolide antibiotic with immunosuppressive properties	[99]
Mycophenolate mofetil	Causes guanosine nucleotides’ depletion in T and B lymphocytes, inhibiting their proliferation	[100]
Cyclosporine	Calcineurin inhibitor with immunomodulatory properties	[101]
Methotrexate	Suppression of inflammation through the inhibition of enzymes responsible for nucleotide synthesis	[102]
Cyclophosphamide	Immunosuppressive agent	[103]
Rituximab	Monoclonal antibody anti-CD20, depleting CD-20 positive cells	[104]
Eculizumab	Monoclonal antibody anti complement C5 protein. Prevent the activation of the complement	[105]
Efgartigimod	IgG1 Fc fragment competing with IgG antibodies	[106]
Duchenne Muscular Dystrophy		
Compound name	Mechanism of action	Reference
Casimersen	Dystrophin gene exon 45 exclusion during splicing	[107]
Viltepso	Dystrophin gene exon 53 exclusion during splicing	[108]
Golodirsen	Dystrophin gene exon 53 exclusion during splicing	[109]
Emflaza	Preserving muscle function	[110]
Eteplirsen	Dystrophin gene exon 51 exclusion during splicing	[111]
Elevidys	Recombinant adeno-associated viral (rAAV) vector-based gene therapy delivering a transgene encoding the micro-dystrophin An engineered protein that retains key functional domains of the wild-type dystrophin	[112]

## 4. Application of Pharmacological Screening to In Vitro NMJ Models

By reconstituting in vitro NMJs affected by neuromuscular disorders, the possibility of screening pharmacological compounds in such systems is appealing. For example, using two-dimensional cell cultures derived from hiPS cells, three different molecules have been discovered and are currently being tested in clinical trials (Table 2). The evolution from simple two-dimensional co-cultures to three-dimensional organoids has involved several research groups. Using an artificial model of NMJ defect by knocking down neural cell adhesion molecules (NCAM) proteins in a two-dimensional co-culture model, Chipman and colleagues tested the ability of four molecules to restore endplate potential [113]. Using a co-culture model with ALS patient cells, kenpaullone was discriminated from a small screening assay comprising valproic acid, isoxazole, lithium chloride (LiCl), and P7C3, an aminopropyl carbazole derivative [16]. In this study, kenpaullone greatly enhanced the outgrowth and branching of neuronal processes, in addition to increasing the survival of MNs for months in laminin-coated plates.

The elegant study carried out by Osaki and colleagues has identified rapamycin and bosutinib as potential molecules to restore contractile function in a microfluidic model of the NMJ from ALS patient-derived components [49]. However, in order to use such models in translational research for the evaluation of pharmacological compounds, the authors raised the question of endothelial cell (EC) permeability, such as the blood–brain barrier. To assess the ability of their compounds to cross this barrier, hIPS-derived ECs were seeded into a reservoir to recreate an artificial endothelial barrier [49]. Under these conditions, treatments with rapamycin and bosutinib failed to increase muscle contractility, supporting the usefulness of having the endothelial barrier in co-culture systems to investigate the potency of compounds. Two years later, the authors again demonstrated the ability of rapamycin and bosutinib to increase muscle contraction in a semi-automated three-dimensional model of the NMJ [114].

As a proof of concept, Santhanam and colleagues used a microfluidic device to assess the ability of co-culture systems to be used for dose–response analysis with curare, α-bungarotoxin, and botulinum neurotoxin type A (BoNTA) derivatives [39]. Using electrical contractions and video recording, dose–response curves were generated, demonstrating the pharmacological relevance of microfluidic systems in establishing a patient-specific platform for screening molecular compounds. Using optogenetics in simple co-culture models, De Lamotte and colleagues evaluated the ability of their system to screen pharmacological compounds [115]. By increasing the concentrations of the botulinum neurotoxin in their model, the authors were able to measure a dose–response effect on calcium flux induced following light stimulations.

Currently, the co-culture systems that have been developed, whether simple co-cultures or microfluidic devices, make it possible to test a small number of molecules by measuring physiological parameters such as neuritic lengths or AChR areas. Interestingly, only a few studies have shown that in vitro NMJ dose–response assays can be performed using electrical or optogenetic stimulations. While these studies demonstrate the potential for in vitro NMJs to be used in large-scale screening assays, numerous challenges such as the automation of these systems remain to be addressed.

**Table 2 ijms-24-15291-t002:** Ongoing clinical trials with molecules identified in hiPS-derived MNs or skeletal muscle.

Amyotrophic Lateral Sclerosis					
Compound name	Mechanism of action	Pluripotent stem cell type	Clinical phase	Clinical trial ID	Reference
Bosutinib	A Src/c-Abl inhibitor	ALS-derived hiPS cells	Phase I	NCT04744532	[116]
Retigabine	A Kv7 channel activator	ALS-derived hiPS cells	Phase II	NCT02450552	[117]
Myotonic dystrophy type 1					
Compound name	Mechanism of action	Pluripotent stem cell type	Clinical phase	Clinical trial ID	Reference
Metformin	AMPK activator	DM1 hES derivatives	Phase III	NCT05532813	[118]

## 5. Discussion

The use of pluripotent stem cells to study the NMJ in vitro has provided unprecedented insights into the interplay between MNs and muscle cells, shedding light on the pathophysiology of various neuromuscular disorders. These studies have evolved from simple two-dimensional models to more sophisticated and complex systems. By differentiating pluripotent stem cells, such as embryonic and induced pluripotent stem cells, into MNs and muscle cells, researchers have partially recapitulated the complex structure and functionality of NMJs in a controlled laboratory environment. These in vitro models have also enabled the investigation in a more human-relevant context. One of the key advantages of using pluripotent stem cells is the ability to derive patient-specific cells from individuals with neuromuscular disorders, allowing for the development of disease-in-a-dish models. These models accurately represent the genetic background of patients and their specific pathophysiological conditions. Studying the NMJs derived from patient-specific pluripotent stem cells provides a unique platform to uncover disease-specific abnormalities in NMJ formation, synaptic connectivity, and functionality. Another major advantage of these models is the reduction in the use of animal models.

In the context of clinical trials, pluripotent stem cell-derived NMJ models hold great promise. They could serve as a platform for high-throughput drug screening, allowing for the rapid testing of different compounds for their ability to restore NMJ function. In addition, these models can contribute to personalised medicine approaches by guiding the development of patient-specific treatments based on the molecular and cellular characteristics of each individual’s disease.

While these systems may offer remarkable potential, many challenges remain. Achieving a high degree of maturation and functionality in vitro, particularly in terms of complex three-dimensional organisation and ultrastructural maturation, is a goal that still requires advances in our understanding of NMJ development. Current co-culture and microfluidic models do not allow for the establishment of three-dimensional NMJ structures, such as the postsynaptic folds in the postsynaptic compartment. This paradigmatic structure is indicative of the good maturation of the synapse. Adding terminal Schwann cells or laminin-rich hydrogels to the culture may help the formation of such complex processes. Throughout the development of hydrogels and bioinks, advances have been made to understand the NMJ microenvironment. However, such systems mean increasing the number of cell types in the synapse, making the process even more complex to automate for large-scale screening assays. In addition, the reproducibility of these matrixes will have to be perfect to enable strong reproducibility between research teams.

While neuromuscular organoids offer exciting opportunities to advance in NMJ research, they are not without challenges. Mimicking the full diversity of cell types present in the neuromuscular niche, such as Schwann cells, glial cells, and satellite cells, provides a very attractive environment for disease modelling. Nevertheless, the complexity of these systems requires the precise control of differentiation protocols, cell ratios, and culture conditions to ensure reproducibility and consistency of results. It is also worth noting that researches in this domain are still in their infancy. The first organoid studies are very recent and need to be further exploited by different research groups. In order to fully exploit the ability of neuromuscular organoids to recapitulate the affected NMJ, the modelling of different neuromuscular pathologies such as SMA or DM1 will also be required. In addition, the evolution of MG models represents an opportunity for the development of new diagnostic tools for patients. Ultimately, the development of robust and automatable assays will be a milestone in the evaluation of pharmacological compounds.

## 6. Conclusions

In conclusion, pluripotent stem cell-derived NMJ models have revolutionised our understanding of neuromuscular diseases and hold tremendous potential for advancing clinical trials. As we continue to improve the accuracy and complexity of these in vitro models, they are expected to play a pivotal role in drug discovery, personalised medicine, and the development of innovative therapies for neuromuscular diseases.

## Figures and Tables

**Figure 1 ijms-24-15291-f001:**
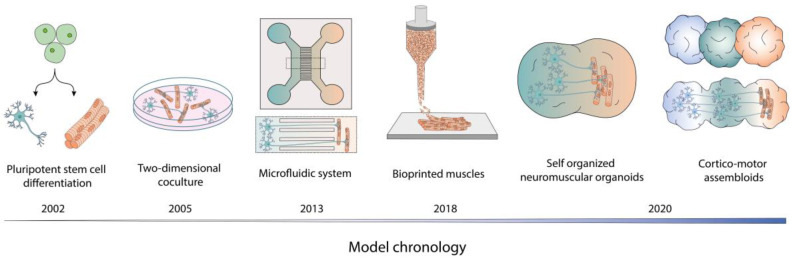
The different established NMJ models derived from pluripotent stem cell differentiation.

## Data Availability

No data were created.

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
