# Peer review of "Unlocking the Complexity of Neuromuscular Diseases: Insights from Human Pluripotent Stem Cell-Derived Neuromuscular Junctions"

_ijms, 2023, doi:10.3390/ijms242015291_

Round 1

Reviewer 1 Report

The manuscript of Gazzola and Martinat is a well-written and comprehensive manuscript on an important topic: human pluripotent stem cell-derived neuromuscular junctions. The authors describe progress in the area over the years and the main breakthroughs, existing challenges, and applications for particular neuromuscular diseases. 

The introduction gives background on neuromuscular diseases and challenges for identifying therapeutics. Further, in a structured way, the authors describe the following milestones chronologically: two-dimensional co-culture, microfluidic, bioprinting, and organoids. Additionally, the authors provide examples of applications in the selected neuromuscular disorders. In the discussion, challenges are highlighted.

The manuscript is logically structured and well-written It is a compendium of up-to-date knowledge on the topic.

Comments: 

1. Are human pluripotent stem cell-derived neuromuscular junctions applied also to model congenital myasthenic syndromes?

2. Please write genes in italics throughout the text (e.g. lines 61-62, SMN1 and SMN2)

3. Line 50 C.elegans should also be in italics

4. I would use the word "pathogenic variants" instead of "mutations"

5. Figure 1: It would be good to write the approximate year on the timeline

Reviewer 2 Report

The authors provide an good review over the past and current development of of models of the NMJ with a focus on neuromuscular disorders. The introduction and description of the models is well described. My major concern is the description of NMJ in different diseases or dieases models. They are rather incomplete or do not have the NMJ as primary focus (i.e. ALS). On the other hand on of the major diseases  (MG) describes mainly NMJ dysfunction than the role of immunological attacks.

Reviewer 3 Report

The focus of the present review is interesting and innovative for the study of the pathogenetic mechanisms underlying neuromuscular diseases. It provides also general knowledge on the in vitro models of NMJ, and on the studies in which these models have been used for drug screening. The use of co-culture and three-dimensional organoids represent interesting models to unravel the pathological mechanism involved in all NMDs, not only in the genetically determined ones, and a short comment on that should be added in the introduction section.

The review is well written and structured, but similar papers have been published on same topic so far, even very recently, and it is not very clear what the present review is adding to those. The authors should at least include them in the reference list, trying also to highlight in the Introduction which are the new aspects that are addressed by the authors in the present paper with respect to what have been published on the same topic so far.

Minor comments:

-       the subtitle of the second paragraph should be changed into “ In vitro NMJ models: from two-dimensional co-culture to tridimensional neuromuscular organoids”.

-       in Table 1 there is no need to include Myotonic Dystrophy type 1 since no FDA approved therapeutics are available so far. This aspect can be annotated in the legend of the same Table.
